## [Peer Review File · EMBO Molecular Medicine]

aMPT and amphetamine ameliorate hyperactivity in a novel mouse model of DTDS

Emma Russo, Ameneh Rezayof, Conner Wallace, Erin Williams, Pieter Beerepoot, Marija Milenkovic, Maria Novalen, Aled Blundell, Tatiana Lipina, Jason Locke, Raveen Christian, Peter Finnie, Landon Edgar, Rachel Tyndale, Dawn Watkins-Chow, Amy Ramsey, Sara Jones, and Ali Salahpour

Corresponding author: Ali Salahpour (ali.salahpour@utoronto.ca)

Review Timeline:

Submission Date:	3rd Apr 25
Editorial Decision:	7th May 25
Revision Received:	5th Nov 25
Editorial Decision:	12th Dec 25
Revision Received:	5th Jan 26
Accepted:	16th Jan 26

Editor: Zeljko Durdevic

Transaction Report:

7th May 2025

Dear Dr. Salahpour,

Thank you for the submission of your manuscript to EMBO Molecular Medicine. We have now received feedback from the three reviewers who agreed to evaluate your manuscript. All three referees recognize interest of the study but also raise serious concerns that should be addressed in a major revision. If you would like to discuss further the points raised by the referees, I am available to do so via email or video. Let me know if you are interested in this option.

We would welcome the submission of a revised version within three months for further consideration. Please let us know if you require longer to complete the revision.

I look forward to receiving your revised manuscript.

Yours sincerely,

Zeljko Durdevic

Zeljko Durdevic
Senior Editor
EMBO Molecular Medicine

We require:

- 1) A .docx formatted version of the manuscript text (including legends for main figures, EV figures and tables). Please make sure that the changes are highlighted to be clearly visible.
- 2) Individual production quality figure files as .eps, .tif, .jpg (one file per figure). For guidance, download the 'Figure Guide PDF': (<https://www.embopress.org/page/journal/17574684/authorguide#figureformat>).
- 3) A .docx formatted letter INCLUDING the reviewers' reports and your detailed point-by-point responses to their comments. As part of the EMBO Press transparent editorial process, the point-by-point response is part of the Review Process File (RPF), which will be published alongside your paper.
- 4) A complete author checklist, which you can download from our author guidelines (<https://www.embopress.org/page/journal/17574684/authorguide#submissionofrevisions>). Please insert information in the checklist that is also reflected in the manuscript. The completed author checklist will also be part of the RPF.
- 5) Please note that all corresponding authors are required to supply an ORCID ID for their name upon submission of a revised manuscript.
- 6) It is mandatory to include a 'Data Availability' section after the Materials and Methods. Before submitting your revision, primary datasets produced in this study need to be deposited in an appropriate public database, and the accession numbers and

database listed under 'Data Availability'. Please remember to provide a reviewer password if the datasets are not yet public (see <https://www.embopress.org/page/journal/17574684/authorguide#dataavailability>).

12) Author contributions: You will be asked to provide CRediT (Contributor Role Taxonomy) terms in the submission system. These replace a narrative author contribution section in the manuscript.

13) A Conflict of Interest statement should be provided in the main text.

14) Every published paper now includes a 'Synopsis' to further enhance discoverability. Synopses are displayed on the journal webpage and are freely accessible to all readers. They include a short stand first (maximum of 300 characters, including space) as well as 2-5 one-sentences bullet points that summarizes the paper. Please write the bullet points to summarize the key NEW findings. They should be designed to be complementary to the abstract - i.e. not repeat the same text. We encourage inclusion of key acronyms and quantitative information (maximum of 30 words / bullet point). Please use the passive voice. Please attach these in a separate file or send them by email, we will incorporate them accordingly.

15) Include a Reagents and Tools Table as part of the Methods section, which can be downloaded from our author guidelines (<https://www.embopress.org/page/journal/17574684/authorguide#structuredmethods>)

***** Reviewer's comments *****

Referee #1 (Comments on Novelty/Model System for Author):

Technical Quality: Solid experimental breadth across biochemistry, behavior, pharmacology, and pharmacokinetics, with generally good statistical reporting. However, some methodological gaps (e.g., group size imbalances) should be clarified. Novelty: The study introduces a new in vivo mammalian model of a DTDS-causing ER-retained DAT variant (A313V), extending prior findings from cellular and *Drosophila* systems. The integration of PK and behavioral pharmacology is a strength. The model is genetically novel and patient-relevant, unlike existing DAT-KO models. The main limitation in novelty lies in that the pathophysiological and pharmacological findings largely reinforce existing knowledge, which restricts the conceptual advance—though the model itself remains a valuable contribution to preclinical DTDS research. Medical Impact: The study addresses an unmet clinical need for DTDS treatment strategies, and the model is promising for future translational work. However, the disease-modifying claims for α MPT are somewhat premature without deeper mechanistic or therapeutic validation.

Referee #1 (Remarks for Author):

This study by Russo et al., titled "Characterization of a novel mouse model of Dopamine Transporter Deficiency Syndrome," presents a novel knock-in (KI) mouse model (DAT-A313V) carrying a pathogenic variant in the dopamine transporter (DAT) gene associated with Dopamine Transporter Deficiency Syndrome (DTDS). Key findings include:

1. In vivo support of in vitro data reporting ER retention of DAT-A313V.
 2. DTDS-relevant changes to dopamine metabolism and behavioral hyperactivity that can be selectively ameliorated in KI mice by amphetamine and α -methyl-para-tyrosine (α MPT) - cooperating data on DAT-KO mice.
 3. A detailed pharmacokinetic analysis of oral noribogaine treatment to contextualize behavioral and biochemical analyses showing reduced locomotor activity in WT and KI mice, but no pharmacochaperoning of DAT in vivo.
- Overall, the study contributes to preclinical understanding of DTDS. The pharmacological data are suggestive of symptomatic treatment strategies, but the disease-modifying claim of α MPT is currently under-supported, and key mechanistic and behavioral questions remain. To merit publication in a high-impact journal such as EMBO Molecular Medicine, the study would benefit from:
- Better assessment of DTDS-relevant dystonic features.
 - Deeper mechanistic insights into the causes of reduced DAT expression.
 - Additional data supporting the disease-modifying effects α MPT data and a more elaborate discussion of potentially conflicting published data

These overall points are elaborated below

Face Validity for DTDS: assessing dystonic features

The authors demonstrate that DAT-A313V mice recapitulate aspects of early DTDS stages, specifically that the mice are hyperactive in the open field, whereas parkinsonian phenotypes in rotarod and gait analysis are absent. It would be relevant to include a targeted evaluation of dystonic features, as these have been reported in DAT-KO mice and are sensitive to gene therapy strategies, making them relevant for assessing translational interventions.

Mechanistic Basis of DAT Reduction

The authors provide compelling evidence for ER retention of the DAT-A313V variant in vivo in the form of a decreased mature/immature DAT ratio in midbrain lysates. Yet, the conclusion that impaired DAT expression is due to ER retention should be supported by additional mechanistic studies, particularly in light of the negative pharmacological chaperoning. Specifically, the authors should assess whether transcriptional regulation of DAT contributes to the changes in expression. Claims of ER retention could also be supported by testing whether inhibitors of ER-associated degradation disproportionately increase immature DAT in KI mice or by demonstrating ER retention through imaging or biochemical strategies (e.g., increased interaction with ER chaperones).

α MPT as a Disease-Modifying treatment strategy

A key finding in this study is that α MPT significantly reduces hyperactivity in DAT-A313V KI mice, as reported previously in DAT-KO mice. While this result is interesting and potentially promising, proposing α MPT as a treatment strategy and potentially a disease-modifying agent requires stronger experimental evidence and a more nuanced discussion of previously published data. The therapeutic and disease-modifying potential of α MPT is based on its ability to reduce hyperactivity and previously reported benefits in DAT-KO mice (e.g., Cyr et al., 2003, where chronic α MPT treatment reduced neurodegenerative phenotypes and

mortality over a 40-week period). However, several studies (including Sotnikova et al.) from the Gainetdinov and Caron groups have shown that α MPT induces acute and severe dopamine depletion in DAT-KO models, leading to profound akinesia, rigidity, and tremor (the so-called "DDD model of Parkinson's disease"). These findings suggest that parkinsonian symptoms in DTDS patients could be worsened by α MPT treatment. Accordingly, the authors should:

A) address the possibility that the observed reduction in hyperactivity reflects general immobility rather than a specific therapeutic effect. Ideally, they should assess whether α MPT-treated KI mice exhibit dystonic or akinetic side effects such as clasping, decreased number of steps, catalepsy, or tremor—or conversely, whether α MPT improves additional aspects of disease beyond hyperactivity suppression (e.g., dystonia, neuronal function, or survival).

B) Include the relevant literature on α MPT-induced parkinsonian phenotypes in DAT-deficient models in the discussion and clinical perspectives.

Another point of concern relates to the large and uneven group sizes in the α MPT treatment experiments. The authors report sample sizes ranging from 8 to 28 per group but do not specify how these sample sizes are distributed across genotypes or treatment conditions. It is currently unclear how the associated statistical analyses are affected by the differences in power to detect treatment effects. If group size imbalance does influence statistical power, the authors should either randomly downsample to achieve more equal group sizes or collect additional data to balance the cohorts. Clear documentation of exact N values for each genotype and treatment condition should be included in the figure legends for all experiments.

To contextualize the α MPT treatment responses and offer mechanistic insight, it would also be relevant to evaluate TH protein expression.

Minor Comments

- First sentence of the abstract appears to be missing the word "for."
- In general, the manuscript includes detailed statistical reporting, but whenever data are presented in parentheses, it is not always specified whether the numbers refer to WT or KI mice. This should be clarified for readability.
- Figures 7C and 7D appear mislabelled; according to the text, α MPT has a significant effect in WT but not A314V.
- Please clarify whether both males and females were used.
- Typo in Figure 7 title: remove extraneous "x."
- Page 10: mention vehicle control explicitly when describing the AMPH treatment (third line from the top) to align the text with the statistical analysis.
- Please clarify the interpretation of Kasture et al., 2016. The current text states: "Interestingly, Kasture et al., 2016 showed that feeding adult *Drosophila* with noribogaine for only 5 days led to 'substantially' lower levels of hDAT G140Q variant in axonal compartments." However, Kasture et al. actually showed that noribogaine increased axonal expression of hDAT G140Q, though the levels remained lower than in WT flies. Please revise this sentence for accuracy.
- The Mir et al. reference appears to be in a different format from the others.
- The Sotnikova et al. reference is cited twice and should be consolidated.

Referee #2 (Remarks for Author):

Russo and collaborators aimed to determine possible treatments for the Dopamine Transporter Deficiency Syndrome (DTDS) that is considered a pharmaco-resistant condition. They developed a novel mouse model for DTDS, harboring the A313V knock-in dopamine (DA) transporter (DAT) variant. The A313V mice are hyperactive and have decreased DA striatal content. They discovered that both alpha-methyl-para-tyrosine (an inhibitor of tyrosine hydroxylase) as well as amphetamine, compounds that are FDA approved, ameliorate the hyperactive nature of this mouse model. This is significant since DTDS is considered a "pharmaco-resistant condition".

Review Points

- 1) In Western blots, the kDa MW should be present.
 - 2) If DA tissue content is significantly lower in the A313V mouse, why is the peak of the electrically stimulated DA release higher?
 - 3) Changes in TH phosphorylation/activity should be evaluated in the A313V mouse.
 - 4) Considering the hyperactive nature of the A313V mouse, in addition to weight, possible changes in body composition should be evaluated (lean vs. fat).
 - 5) Some figures are missing statistical symbols (e.g. 7d).
 - 6) The comparison between cell and animal studies about noribogaine results needs further clarification. Multiple cell studies adopted biotinylation and/or other techniques to determine changes in cell surface expression of the DAT. Simple western blot analysis cannot determine these changes.
- Nonetheless, this is a very elegant study.

Referee #3 (Comments on Novelty/Model System for Author):

Overall, the authors have thoroughly planned and characterized the novel mouse model for Dopamine Transporter Deficiency Syndrome (DTDS). However, several aspects require further clarification or discussion:

1. The reduced tissue levels of dopamine (DA) detected in the A313V mice suggest a diminished releasable pool of DA (Figure 3A). However, fast-scan cyclic voltammetry recordings showed significantly higher peaks during DA clearance measurements in the mutants. The authors should discuss this discrepancy further.
2. The pharmacological chaperone (noribogaine) presumably binds specifically to immature DAT. However, behavioral data demonstrated reduced locomotor activity in both WT and A313V mutant mice post-treatment (Figure 8C & D). Given the higher proportion of immature DAT in A313V mice (presumably misfolded and retained in the ER, Figure 1E), it remains unclear why noribogaine affects locomotion similarly in WT mice.
3. It would also be beneficial for the authors to confirm explicitly that the pharmacological chaperone (noribogaine) used in this study indeed binds immature DAT, possibly through a Western blot analysis of striatal and midbrain DAT post-treatment.
4. A relatively high dose of amphetamine (AMPH, 3 mg/kg) was used to evaluate its effect on relieving the hyperlocomotion in mutant mice. However, Figure 6A indicates that AMPH treatment in WT mice did not substantially increase locomotion compared to their baseline during the first 60 minutes of habituation. The authors should clarify or further discuss this observation.
5. The authors claim increased sensitivity of A313V mutants to 250 mg/kg alpha-methyl-para-tyrosine (αMPT; Figure 7 A&B). However, given the already low baseline locomotion of WT mice (Figure 7A), a potential floor effect could prevent detecting further reductions due to αMPT treatment. Further discussion of this possibility is needed.
6. Notably, when examining dose-dependent effects of αMPT, the A313V mice did not display baseline hyperlocomotion relative to WT under vehicle conditions (Figure 7 C&D). The authors need to clarify whether this is an error in labeling or data presentation.
7. Figures and figure legends do not consistently match. For example, there are two separate figures depicting dose-dependent αMPT treatments but only one corresponding figure legend (Figure 7).
8. CRISPR-Cas9 genome editing is known to have off-target effects. To ensure specificity, the authors should compare sequences of the DAT exon carefully, confirming the presence of only the intended mutation.

Dear Dr. Durdevic

We thank the reviewers for their thorough, constructive, and positive comments. Below we have addressed each reviewer comment point by point (our answers in blue font). We have conducted a number of new experiments to fully address the reviewer's comments (full list of all edits is at the bottom of the current document). We believe that the revised version, including the recommendations of the reviewers and including all the new experiments, is substantially improved and we hope that the new results and our new additions/discussions fully address the concerns of the referees and make the manuscript suitable for publication. Thank you for your consideration.

Ali Salahpour, PhD,
Professor and Chair.

Referee #1 (Comments on Novelty/Model System for Author):

Technical Quality: Solid experimental breadth across biochemistry, behavior, pharmacology, and pharmacokinetics, with generally good statistical reporting. However, some methodological gaps (e.g., group size imbalances) should be clarified.

Novelty: The study introduces a new in vivo mammalian model of a DTDS-causing ER-retained DAT variant (A313V), extending prior findings from cellular and *Drosophila* systems. The integration of PK and behavioral pharmacology is a strength. The model is genetically novel and patient-relevant, unlike existing DAT-KO models. The main limitation in novelty lies in that the pathophysiological and pharmacological findings largely reinforce existing knowledge, which restricts the conceptual advance-though the model itself remains a valuable contribution to preclinical DTDS research.

Medical Impact: The study addresses an unmet clinical need for DTDS treatment strategies, and the model is promising for future translational work. However, the disease-modifying claims for α MPT are somewhat premature without deeper mechanistic or therapeutic validation.

Referee #1 (Remarks for Author):

This study by Russo et al., titled "Characterization of a novel mouse model of Dopamine Transporter Deficiency Syndrome," presents a novel knock-in (KI) mouse model (DAT-A313V) carrying a pathogenic variant in the dopamine transporter (DAT) gene associated with Dopamine Transporter Deficiency Syndrome (DTDS). Key findings include:

1. In vivo support of in vitro data reporting ER retention of DAT-A313V.
2. DTDS-relevant changes to dopamine metabolism and behavioral hyperactivity that can be selectively ameliorated in KI mice by amphetamine and α -methyl-para-tyrosine (α MPT) - cooperating data on DAT-KO mice.
3. A detailed pharmacokinetic analysis of oral noribogaine treatment to contextualize behavioral and biochemical analyses showing reduced locomotor activity in WT and KI mice, but no pharmacochaperoning of DAT in vivo.

Overall, the study contributes to preclinical understanding of DTDS. The pharmacological data are suggestive of symptomatic treatment strategies, but the disease-modifying claim of α MPT is currently under-supported, and key mechanistic and behavioral questions remain. To merit publication in a high-impact journal such as EMBO Molecular Medicine, the study would benefit from:

- Better assessment of DTDS-relevant dystonic features.
- Deeper mechanistic insights into the causes of reduced DAT expression.
- Additional data supporting the disease-modifying effects α MPT data and a more elaborate discussion of potentially conflicting published data

These overall points are elaborated below

Face Validity for DTDS: assessing dystonic features

The authors demonstrate that DAT-A313V mice recapitulate aspects of early DTDS stages, specifically that the mice are hyperactive in the open field, whereas parkinsonian phenotypes in rotarod and gait analysis are absent. It would be relevant to include a targeted evaluation of dystonic features, as these have been reported in DAT-KO mice and are sensitive to gene therapy strategies, making them relevant for assessing translational interventions.

Thank you for this suggestion. To this end, we have performed the pole reversal test and the foot fault test in WT, A313V, and DKO mice as a control group as suggested by the referee. These are the same tests for dystonia that were used in Ng et al., 2022 cited by the referee. In Ng et al., 2022, the authors used DAT-KO mice as a model of DTDS and used pole reversal and foot fault as a measure of dystonia. As shown below, we do not observe any significant differences in the pole reversal test between WT, A313V or DKO mice in either time to turn (T) or latency to descend the pole. We also did not observe any differences in % foot faults between WT, DKO, and A313V mice.

This is unlike Ng et al., 2022 that did see changes in the DKO mice, which we are unable to replicate under our experimental conditions.

To expand our measures of dystonia, we also assessed clasping in WT, A313V, and DKO mice. This test has been used as a measure of dystonia (Liu et al., 2016; Pappas et al., 2015). Again, in our experimental conditions we do not observe any differences in clasping between WT, A313V, and DAT-KO mice (see below). In sum, under our housing and experimental conditions, neither DKO nor A313V display features of dystonia. We have included these new results to address the referee comment above.

However, we do not feel that this inability to detect dystonia in these animals is surprising. Rather, several studies have reported that it is difficult to observe behavioral manifestations of parkinsonism/dystonia as a result of insult to the dopamine system in mice. For example, and as discussed in our manuscript, Golden et al., 2013 showed that developmental ablation of approximately 90% of ventral tegmental area and substantia pars compact dopaminergic neurons lead to no observable motor phenotype. One of the behaviors reported in Golden et al 2013 was the pole reversal test, and, likewise, there was no difference in time to T turn nor latency to descend in that study.

To summarize, we tried to replicate the results reported by Ng et al, 2022, and under our experimental conditions (and potentially housing conditions, different animal facility, etc), we are unable to observe any changes reflecting dystonia between WT, A313V or DKO. Since these are mostly 'negative' results, which could be due to different experimental and housing conditions, we feel it is best not to include them in the main manuscript, and prefer to only show them here for the referees and editors.

Mechanistic Basis of DAT Reduction

The authors provide compelling evidence for ER retention of the DAT-A313V variant *in vivo* in the form of a decreased mature/immature DAT ratio in midbrain lysates. Yet, the conclusion that impaired DAT expression is due to ER retention should be supported by additional mechanistic studies, particularly in light of the negative pharmacological chaperoning. Specifically, the authors should assess whether transcriptional regulation of DAT contributes to the changes in expression. Claims of ER retention could also be supported by testing whether inhibitors of ER-associated degradation disproportionately increase immature DAT in KI mice or by demonstrating ER retention through imaging or

biochemical strategies (e.g., increased interaction with ER chaperones).

We appreciate the recommendation to analyze transcriptional regulation of DAT in WT and A313V mice. As such, we have performed rt-qPCR on midbrain samples in these mice. We did indeed observe reduction in DAT mRNA in the A313V mice (see new Figure 3f) suggesting that in part, reductions in DAT protein are also due to reductions in transcriptional regulation of DAT in the A313V mice. With regard to ER localization, our results clearly show that EndoH reduces the molecular weight of the immature DAT protein species seen in the midbrain of WT mice, a finding that confirms that there is indeed DAT retained in the ER. Nonetheless, the reduction in DAT mRNA is a significant finding with important implications for the noribogaine experiments. As such, we have added discussion of these findings to the manuscript and the possibility that the lack of DAT rescue by noribogaine could be partially due to decreased DAT mRNA in the A313V mice. Furthermore, we assessed transcriptional regulation of TH and VMAT2, and found that A313V mice have decreased midbrain mRNA of both for VMAT2 and TH (new Figure 2e, and new EV Figure 1c).

To address the reviewer comment regarding testing inhibitors of ER-associated degradation, we have performed an experiment wherein pifithrin-u, an inhibitor of HSP70, was administered over the course of 5 days to WT and A313V mice. Pifithrin-u has previously been shown to increase the levels of DTDS DAT variants in drosophila, recapitulating the actions of noribogaine (Kasture et al., 2016, Asjad et al., 2017). In new Fig 13, we show that Pifithrin-u treatment has no effect on WT DAT levels but interestingly the 5-day treatment resulted in a very significant increase in mature A313V DAT levels ONLY in the midbrain with no changes in A313V DAT levels in the striatum (Insert new Fig Number here). This is a very interesting result and partially recapitulates previous published results with Pifithrin-u in drosophila. But intriguingly, in our paradigm of 5-day treatment, we only saw an increase in midbrain DAT levels and not in the striatum. Accordingly, we also didn't observe any motor improvement with Pifithrin-u. It is possible that our 5-day regimen was too short for the newly folded DAT to migrate from midbrain to the striatum and this is why we only see increases in midbrain and no motor improvements. In future studies, time-course experiments could be conducted (beyond the scope of the current paper) to see if longer treatment with Pifithrin-u can upregulate A313V levels in the striatum as well and improve motor symptoms. We discuss these findings in detail in the discussion section.

αMPT as a Disease-Modifying treatment strategy

A key finding in this study is that αMPT significantly reduces hyperactivity in DAT-A313V

KI mice, as reported previously in DAT-KO mice. While this result is interesting and potentially promising, proposing α MPT as a treatment strategy and potentially a disease-modifying agent requires stronger experimental evidence and a more nuanced discussion of previously published data. The therapeutic and disease-modifying potential of α MPT is based on its ability to reduce hyperactivity and previously reported benefits in DAT-KO mice (e.g., Cyr et al., 2003, where chronic α MPT treatment reduced neurodegenerative phenotypes and mortality over a 40-week period). However, several studies (including Sotnikova et al.) from the Gainetdinov and Caron groups have shown that α MPT induces acute and severe dopamine depletion in DAT-KO models, leading to profound akinesia, rigidity, and tremor (the so-called "DDD model of Parkinson's disease"). These findings suggest that parkinsonian symptoms in DTDS patients could be worsened by α MPT treatment. Accordingly, the authors should:

A) address the possibility that the observed reduction in hyperactivity reflects general immobility rather than a specific therapeutic effect. Ideally, they should assess whether α MPT-treated KI mice exhibit dystonic or akinetic side effects such as clapping, decreased number of steps, catalepsy, or tremor-or conversely, whether α MPT improves additional aspects of disease beyond hyperactivity suppression (e.g., dystonia, neuronal function, or survival).

B) Include the relevant literature on α MPT-induced parkinsonian phenotypes in DAT-deficient models in the discussion and clinical perspectives.

Thank you for your comments. Indeed, high doses of α MPT lead to severe dopamine depletion in DAT-KO mice and consequent akinesia, rigidity, and tremor. However, we would like to remind the reviewer that in the A313V animals, we do not observe severe akinesia, rigidity, or tremor after α MPT treatment, we only see a reduction in locomotor activity while animals retain normal ambulation (See figure below):

The reduced sensitivity of A313V animals to aMPT is due to the fact that there is still some dopamine uptake in A313V mice and dopamine tissue content in A313V mice is about 20% of WT levels as compared to DAT-KO which only have 2-5% of WT dopamine tissue levels and are highly sensitive to aMPT mediated dopamine depletion.

Nevertheless, to address this important concern of the referee and to expand the potential for aMPT to treat all DTDS patients, even those with full DAT deficiency (like a DAT-KO), we have performed dose response experiments of aMPT in DAT-KO mice, a model of “extreme” DTDS (zero DAT expression).

Our new results show that aMPT dose dependently inhibits DAT-KO hyperactivity with the higher doses (250, 125, 62.5mg/kg) inducing complete akinesia and rigidity. Importantly however, with lower doses (7, 15.5, 31 mg/kg), hyperactivity of the animals is reduced with low levels of akinesia and in fact, 7 mg/kg of aMPT does not induce any akinesia at all but significantly reduces hyperactivity. These results show that from a clinical perspective, it may be possible to find an aMPT dose that would have clinical

benefits to reduce the hyperactivity of DTDS in infants without inducing akinesia and rigidity as a major side effect as noted by the referee. We feel that these new additional dose response data further supported translational potential of aMPT as a treatment for early stage DTDS. This exciting new data is shown in Fig 10 in the resubmission. We have also fully discussed this including the potential dystonic/parkinsonian adverse effects that could accompany aMPT to cover the literature as suggested by the referee.

To supplement this finding, we have also recorded a video demonstrating the lack of akinesia and rigidity in DAT-KO in response to 7mg/kg aMPT. The link to the video can be found here: <https://doi.org/10.6084/m9.figshare.30209422.v1>. We hope that this demonstrates the lack of akinesia in response to 7 mg/kg aMPT in DAT-KO mice, and demonstrates proof of principle for using aMPT to treat early-stage DTDS.

Another point of concern relates to the large and uneven group sizes in the α MPT

treatment experiments. The authors report sample sizes ranging from 8 to 28 per group but do not specify how these sample sizes are distributed across genotypes or treatment conditions. It is currently unclear how the associated statistical analyses are affected by the differences in power to detect treatment effects. If group size imbalance does influence statistical power, the authors should either randomly downsample to achieve more equal group sizes or collect additional data to balance the cohorts.

Thank you for this comment. In our methods section, we clarify that the vehicle animals were combined (n=26-28) while drug tested animals were between N of 3-9. This was already stated in the methods section, but we have now included these details in the figure legends. We hope that this clarifies the different N between vehicle and drug-treated for the referee.

Clear documentation of exact N values for each genotype and treatment condition should be included in the figure legends for all experiments.

Thank you; we have included details of exact N values for each genotype and treatment condition for all figure legends.

To contextualize the α MPT treatment responses and offer mechanistic insight, it would also be relevant to evaluate TH protein expression.

We agree that this is an important addition to our manuscript. We have performed a western blot to assess the levels of TH and phospho-TH s40 in the A313V and WT mice. Please see Fig. 2a-d as new results showing that TH and pTH levels are also reduced in A313V mice as a potential compensation mechanism due to hyperdopaminergic state, similar to what has been reported in DAT-KO animals.

Minor Comments

- First sentence of the abstract appears to be missing the word "for."

Thank you, we have added the missing word.

- In general, the manuscript includes detailed statistical reporting, but whenever data are presented in parentheses, it is not always specified whether the numbers refer to WT or KI mice. This should be clarified for readability.

We have gone through and clarified which data in parentheses is associated with which genotype and/or treatment.

- Figures 7C and 7D appear mislabelled; according to the text, α MPT has a significant effect in WT but not A314V.

Thank you for catching this, indeed, the figures were mislabeled and have been corrected.

- Please clarify whether both males and females were used.

We have clarified in the figure legends the number of males and females used in each experiment.

- Typo in Figure 7 title: remove extraneous "x."

Thank you, we have removed the erroneous x.

- Page 10: mention vehicle control explicitly when describing the AMPH treatment (third line from the top) to align the text with the statistical analysis.

We now mention the vehicle control explicitly when discussing the results of the AMPH experiment. The description of the statistics in the text is now as follows: A two-way ANOVA was performed on the sum of total distance traveled post-treatment in WT and A313V mice. There were no main effects of genotype ($F_{1,17}=0.3093$, $p=0.5853$) or treatment ($F_{1,17}=0.06423$, $p=0.8030$), but there was a genotype by treatment interaction ($F_{1,17}=22.38$, $p=0.0002$). Šidák's multiple comparisons within genotype revealed a significant difference between WT mice treated with vehicle or AMPH and between A313V mice treated with vehicle or AMPH. WT mice treated with 3 mg/kg AMPH showed an eightfold increase in locomotor activity compared to vehicle-treated WT animals (vehicle: 378.75 ± 64.65 cm vs 3mg/kg AMPH: 3190.20 ± 693.61 cm, $p=0.0171$) (Fig. 8a,c). However, A313V mice treated with 3 mg/kg AMPH traveled approximately six times less than A313V mice that received vehicle (vehicle: 3698.60 ± 1052.91 cm vs 3mg/kg AMPH: 568.86 ± 216.98 cm, $p=0.0029$) (Fig. 8b,c).

- Please clarify the interpretation of Kasture et al., 2016. The current text states: "Interestingly, Kasture et al., 2016 showed that feeding adult *Drosophila* with noribogaine for only 5 days led to 'substantially' lower levels of hDAT G140Q variant in

axonal compartments." However, Kasture et al. actually showed that noribogaine increased axonal expression of hDAT G140Q, though the levels remained lower than in WT flies. Please revise this sentence for accuracy.

Thank you for suggesting that we clarify and reconsider our interpretation. Indeed, Kasture et al. showed that feeding adult flies for 5 days increased hDAT G140Q in axonal compartments as well as when flies were fed with noribogaine beginning in the instar larval phase. We have reworded this portion of our discussion as follows: "What remains puzzling is that Asjad et al., 2017 demonstrated the ability to detect expression of human DAT G140Q or V158F variants in axonal compartments of drosophila when flies were fed with noribogaine beginning in the instar larval stage throughout adulthood. Interestingly, Kasture et al., 2016 showed that feeding adult drosophila with noribogaine for only 5 days was also sufficient to achieve this effect. However, in our studies, where noribogaine was administered to the A313V mice only in adulthood, rescue was not achieved. Perhaps treatment with noribogaine needs to begin earlier in life, and for a longer duration, to observe DAT rescue in mice. However, this is difficult to experimentally verify because noribogaine levels in the brain fall below the efficacious 100 μ M levels after approximately 84 hours of administration, despite repeated dosing every 48 hours. Therefore, it may not be pharmacokinetically feasible to conduct prolonged noribogaine studies with early intervention in mice."

- The Mir et al. reference appears to be in a different format from the others.

We have corrected the format of this citation.

- The Sotnikova et al. reference is cited twice and should be consolidated.

We have removed the duplicate Sotnikova reference.

Referee #2 (Remarks for Author):

Russo and collaborators aimed to determine possible treatments for the Dopamine Transporter Deficiency Syndrome (DTDS) that is considered a pharmaco-resistant condition. They developed a novel mouse model for DTDS, harboring the A313V knock-in dopamine (DA) transporter (DAT) variant. The A313V mice are hyperactive and have decreased DA striatal content. They discovered that both alpha-methyl-para-tyrosine (an inhibitor of tyrosine hydroxylase) as well as amphetamine, compounds that are FDA approved, ameliorate the hyperactive nature of this mouse model. This is significant

since DTDS is considered a "pharmaco-resistant condition".

Review Points

1) In Western blots, the kDa MW should be present.

Thank you, we have now added kDa MW to all western images.

2) If DA tissue content is significantly lower in the A313V mouse, why is the peak of the electrically stimulated DA release higher?

In FSCV measurements, peak height of the DA signal correlates more with uptake ability than dopamine tissue content. For example, when measuring DA release in vivo with FSCV, one of the main effects observed with DAT blockers is increased peak of the DA signal which could be interpreted as 'increased' release but in reality this is due to the fact that DA remains longer in the extracellular space and can diffuse farther to reach FSCV electrode (see example of data below from Hersey et al., 2016).

Therefore, the moderate (~30% increase in peak height) seen in the A313V mice is consistent with similar or modestly reduced release of DA coupled with impaired uptake compared to WT. Indeed, what this result shows is that there is an increase in Peak Height but not an increase in "release". Also, the decrease in tissue content seen in A313V mice (20% of WT) is not as drastic as in DAT KO mice (2-5% of WT), which could further explain why the 'peak height' of A313V is not reduced while the 'peak height' of DAT-KO is reduced. Note that although DAT-KO tissue content is 2-5% of WT, 'Peak height' of DA release for these animals is ONLY reduced to 25% of WT, again demonstrating that the lack of uptake has a major effect on DA 'Peak height' as measured by FSCV and that peak height is not only determined by DA release. Lastly, our results with A313V are not the first to demonstrate a 'discrepancy' between tissue content and DA-Release/Peak height in FSCV. Indeed, Sorensen et al, 2021 showed that in AAA-DAT mice (lacking the PDZ sequence of DAT), DA tissue content is down by more than 50% in AAA-DAT mice vs WT, but DA peak height is higher by ~50% for the AAA-DAT mice (see results from Sorensen et al 2021, below). This is very similar to what we have observed in our A313V mice vs WT.

In sum, peak-height measurements of DA release by FSCV is dependent on both DA tissue content but also DAT uptake capacity and although in some cases, tissue content might be lower, it may not result in reduced peak-height release as measured by FSCV if DAT uptake activity is also reduced. This is now fully discussed in the revised manuscript.

Hersey et al., 2016

3) Changes in TH phosphorylation/activity should be evaluated in the A313V mouse.

Thank you for this suggestion; this was also suggested by reviewer 1. We have performed western blots to assess TH and pTH s40 levels, seen in Figure 2, and show that indeed TH and pTH levels are reduced in A313V animals compared to WT,

4) Considering the hyperactive nature of the A313V mouse, in addition to weight, possible changes in body composition should be evaluated (lean vs. fat).

We agree that this would add insightful information regarding the A313V mice. Given this, we have examined body composition using a DEXA scan to assess bone mineral content/weight, lean mass/body weight, fat mass/body weight, and bone area. This new

data is now presented Figure 5 in the manuscript. Our results show that A313V mice have reduced lean mass, fat mass, and decreased bone area compared to WT mice; the body composition of A313V mice is more similar to that of DKO mice.

5) Some figures are missing statistical symbols (e.g. 7d).

Thank you for this comment; we have chosen to not include statistical symbols when no significance is present. We believe confusion regarding 7d specifically may have arisen because the graphs were labeled incorrectly. To clarify, we have added “ ns $p < 0.05$ (not shown)” in figure legends where applicable.

6) The comparison between cell and animal studies about noribogaine results needs further clarification. Multiple cell studies adopted biotinylation and/or other techniques to determine changes in cell surface expression of the DAT. Simple western blot analysis cannot determine these changes.

We appreciate this comment-to address the referee's concern, we have expanded the discussion to include the limitation that western blotting may not be sensitive enough to detect minor upregulation of DAT by noribogaine, and future studies could benefit from using techniques such as DA uptake to assess noribogaine mediated rescue of DAT.

Referee #3 (Comments on Novelty/Model System for Author):

Overall, the authors have thoroughly planned and characterized the novel mouse model for Dopamine Transporter Deficiency Syndrome (DTDS). However, several aspects require further clarification or discussion:

1. The reduced tissue levels of dopamine (DA) detected in the A313V mice suggest a diminished releasable pool of DA (Figure 3A). However, fast-scan cyclic voltammetry recordings showed significantly higher peaks during DA clearance measurements in the mutants. The authors should discuss this discrepancy further.

Please see answer to point #2 of Ref 2 above, who also asked this question.

2. The pharmacological chaperone (noribogaine) presumably binds specifically to immature DAT. However, behavioral data demonstrated reduced locomotor activity in both WT and A313V mutant mice post-treatment (Figure 8C & D). Given the higher proportion of immature DAT in A313V mice (presumably misfolded and retained in the

ER, Figure 1E), it remains unclear why noribogaine affects locomotion similarly in WT mice.

Thank you for this comment. Noribogaine is known to bind to a number of other pharmacological targets in addition to DAT. For example, noribogaine binds to SERT (Bhat et al., 2021; El-Kasaby et al., 2024; El-Kasaby et al., 2010), opioid receptors, NMDA receptors, and nicotinic receptors (a review by Ona et al., 2023). Therefore, the decrease in locomotor activity seen in WT mice after administration of noribogaine may be due to several other factors unrelated to DAT. To demonstrate its general effects of sedation and therefore decreases in locomotor activity, we administered ibogaine (parent compound/Noribogaine is the active metabolite of ibogaine) or vehicle to WT mice and assessed their locomotor activity. As seen below, WT mice given 100mg/kg ibogaine via oral gavage (n=8) traveled significantly less distance over the course of 60 minutes in the open field test compared to WT animals treated with vehicle (n=8). We believe this data demonstrates the general locomotor-reducing activity of ibogaine in WT mice, and explains the reduction in locomotor activity seen in Figure 8. We have added a sentence in the manuscript commenting on the potential sedating effect of noribogaine that may explain the reduction in locomotor activity even if we don't see an effect of A313V DAT protein expression. The results below are included here for referee's appraisal

3. It would also be beneficial for the authors to confirm explicitly that the pharmacological chaperone (noribogaine) used in this study indeed binds immature DAT, possibly through a Western blot analysis of striatal and midbrain DAT post-treatment.

Thank you for your comment. However, we are uncertain what kind of experimental conditions would allow us to demonstrate this request. To our knowledge, it is not possible to demonstrate a drug-protein interaction through western blotting. Figure 9 in the manuscript depicts western blots performed to assess the levels of DAT in the midbrain and striatum of WT and A313V mice after administration of noribogaine, where we did not see changes in the levels of DAT post-treatment.

Furthermore, we have performed extensive PK studies to ensure we have reached the efficacious dose of 30uM and above for pharmacological chaperone activity at DAT by noribogaine (Beerepoot et al., 2016, Sutton et al., 2022, Bhat et al., 2020, Kasture et al., 2016). We confirmed we had achieved the dose needed for chaperone activity at DAT, but did not detect such.

4. A relatively high dose of amphetamine (AMPH, 3 mg/kg) was used to evaluate its effect on relieving the hyperlocomotion in mutant mice. However, Figure 6A indicates that AMPH treatment in WT mice did not substantially increase locomotion compared to their baseline during the first 60 minutes of habituation. The authors should clarify or further discuss this observation.

Figure 6A shows 60 minutes of habituation and baseline locomotor activity of WT mice prior to treatment with 3 mg/kg AMPH or vehicle. Treatment was performed at minute 60, marked by an arrow in the figure. After administration of 3 mg/kg AMPH, from minutes 65-150, WT mice displayed significantly increased levels of locomotor activity as measured by an increase in total distance traveled compared to WT mice that received vehicle (This is shown as a bar graph in Figure 6B). As expected by the referee, we did indeed observe an increase in locomotor activity in WT mice in response to administration of 3 mg/kg AMPH compared to vehicle-treated WT mice. Hopefully our explanation above clarifies these observations for the referee.

5. The authors claim increased sensitivity of A313V mutants to 250 mg/kg alpha-methyl-para-tyrosine (αMPT; Figure 7 A&B). However, given the already low baseline locomotion of WT mice (Figure 7A), a potential floor effect could prevent detecting further reductions due to αMPT treatment. Further discussion of this possibility is needed.

Thank you for this comment. Indeed, we cannot rule out the possibility that 250 mg/kg αMPT cannot substantially decrease the levels of locomotor activity in WT mice due to lower baseline levels. However, we believe that this is beyond the scope of this study, as the primary goal is to demonstrate the ability of αMPT to reduce the hyperactivity of the A313V mice as a potential treatment for the initial stages of DTDS. However, Sotnikova et al., 2005 measured the levels of extracellular striatal dopamine in WT and

DKO mice after administration of 250mg/kg aMPT (see Figure 1 from Sotnikova et al 2005 below). Although 250 mg/kg aMPT reduced the levels of striatal extracellular DA in WT mice, the reduction was only ~50% of that of baseline levels in WT mice. In contrast, DKO mice treated with 250mg/kg aMPT showed a near complete depletion of extracellular striatal DA. Based on this evidence, it is unlikely that the WT mice are displaying a floor effect in response to aMPT. Nonetheless, the goal of the aMPT experiments in the A313V mice is to demonstrate a potential treatment for the symptoms of DTDS, and not to primarily suggest they are more sensitive to aMPT than WT mice.

6. Notably, when examining dose-dependent effects of aMPT, the A313V mice did not display baseline hyperlocomotion relative to WT under vehicle conditions (Figure 7 C&D). The authors need to clarify whether this is an error in labeling or data presentation.

Thank you for bringing this to our attention. Indeed, the graphs C and D were mislabeled. 7C is now corrected to be the WT bar graph and 7D is now corrected to be the A313V graph; originally these were swapped. Thank you for noting this which has now been corrected.

7. Figures and figure legends do not consistently match. For example, there are two separate figures depicting dose-dependent aMPT treatments but only one corresponding figure legend (Figure 7).

We believe a second figure 7 had been submitted in error. Upon resubmission, we confirm that there is only one corresponding figure legend for each figure.

8. CRISPR-Cas9 genome editing is known to have off-target effects. To ensure specificity, the authors should compare sequences of the DAT exon carefully, confirming the presence of only the intended mutation.

We have sequenced the entire coding region of the A313V mDAT gene and confirm that there are no off target effects caused by the CRISPR-Cas9 editing. The link to the relevant files can be found here: <https://osf.io/vmupw/> . This link has now been added to the manuscript.

Summary of new additions to the revised version.

- Measured dystonia in WT, A313V, and DKO through the pole reversal test, the foot fault test, and clasping. Results shown in response to ref document.
- Measured transcriptional regulation of DAT, TH, and VMAT in WT, A313V, and DKO mice (new Figure 3f, new Figure 2e, and new EV Figure 1c)
- Performed a chronic pifithrin-u experiment to assess if inhibition of ER-associated degradation could lead to increased levels of DAT in the A313V mice (new Figure 12, 13). We discuss these findings in detail in the discussion
- To address the important concern that aMPT may lead to worsening of parkinsonism-dystonia in patients with DTDS, we administered a series of doses of aMPT to DAT-KO mice, who fully lack DAT expression (new figure 9)
- Recorded a video demonstrating lack of rigidity and akinesia in DAT-KO mice in response to 7mg/kg aMPT
- Included in figure legends all N values for genotype, sex, and treatment condition
- Performed a western blot to assess the levels of TH and pTHs40 in WT and A313V mice (new figure 2)
- Clarified and reconsidered our discussion of the Kasture et al., 2016 experiment

- Added kDA MW to all western images
- Added a detailed discussion of the peak-height of electrically stimulated DA release, as measured by FSCV, being higher in A313V mice compared to WT mice
- Examined body composition including lean mass, fat mass, and bone area in WT, A313V, and DKO mice (new figure 5)
- Clarified the choice to not include statistical symbols for non-significant results in the figures and added “ ns $p < 0.05$ (not shown)” in figure legends where applicable
- Expanded the discussion to address the concern that western blots may not be sensitive enough to detect minor up regulation of DAT by noribogaine
- Added a sentence to discuss the potential sedative effect of noribogaine to address the concern that noribogaine decreased locomotor activity in both WT and A313V mice, and performed an experiment where WT mice were given 100mg/kg of ibogaine (of which noribogaine is the active metabolite) for 60 minutes to show that in WT mice, ibogaine decreases locomotor activity. Results shown in response to ref document.
- Sequenced the entire coding region of the A313V mDAT gene to confirm there are no off target effects caused by CRISPR-cas9 editing
- Corrected minor typos, duplicate references, mislabeling, corrected citation formatting

12th Dec 2025

Dear Dr. Salahpour,

Thank you for the submission of your revised manuscript to EMBO Molecular Medicine. I am pleased to inform you that we will be able to accept your manuscript pending the following final amendments:

- 1) Please implement the referee #1 suggestions.
- 2) Please consider changing the current title to "Alpha-methyl-para-tyrosine and amphetamine ameliorate hyperactivity in a novel mouse model of Dopamine Transporter Deficiency Syndrome"
- 3) Figures: Please reduce number of figures by merging 2-3 current figures, e.g. Fig1 and 2, Fig3, 4 and 5, Fig6 and 7, Fig8 and 9, Fig10 and 11 and, Fig12 and 13. Please note that all figures should fit one page. Please check "Author Guidelines" for more information:

<https://www.embopress.org/page/journal/17574684/authorguide#figureformat>

4) Author checklist: Please submit a complete checklist. <https://www.embopress.org/pb-assets/embo-site/EMBO%20Press%20Author%20Checklist-1642513524327.xlsx>

5) In the main manuscript file, please do the following:

- Please correct the citations of the EV figures to "Figure EV1" etc. and add callouts for Fig 2A, Fig 6F. Figures should be called out in a sequential order. Currently Fig EV5B is called out before Fig EV4B, please correct. Also, there are callouts for Fig. EV8d and Fig EV6 which do not exist, please correct.

- Remove data not shown on page 26 and in figure legends Fig 2,5,6, 7,8,9,11,13.

- Please include structured Methods section that includes a Reagents and Tools Table (should be uploaded as a separate file) followed by a Methods and Protocols section. More information on how to adhere to this format as well as downloadable templates (.docx) for the Reagents and Tools Table can be found in our author guidelines:

<https://www.embopress.org/page/journal/17574684/authorguide#structuredmethods>

An example of a paper with Structured Methods can be found here:

<https://www.embopress.org/doi/full/10.1038/s44320-024-00037-6#sec-4>

- Indicate in legends exact n and exact p values, not a range, along with the statistical test used. To keep the figures "clear" some authors found providing an Appendix table Sx with all exact p-values preferable. You are welcome to do this if you want to.

- Rename "Conflict of interest" to "Disclosure and competing interests statement". We updated our journal's competing interests policy in January 2022 and request authors to consider both actual and perceived competing interests. Please review the policy <https://www.embopress.org/competing-interests> and update your competing interests if necessary.

- In data availability statement replace current text with "This study includes no data deposited in external repositories."

- Merge Funding with Acknowledgement.

- Please correct the reference citation in the reference list. Citations should be listed in alphabetical order. Currently Mir, A., et al is after Aguilar, J. I., et al. Where there are more than 10 authors on a paper, 10 will be listed, followed by "et al.". Also, please remove DOIs. DOIs should only be used for preprints and datasets that have not been published. Please check "Author Guidelines" for more information.

<https://www.embopress.org/page/journal/17574684/authorguide#referencesformat>

6) Appendix: Please move Appendix Methods to main Methods section. Add page numbers to table of contents and correct to "Appendix Figure S1" etc. and "Appendix Table S1".

7) Source data:

- Please upload high-resolution source data images for blots in Fig1 and 2. Currently these are pixelated.

- Correct panel labeling for Fig12 source data to b, c, d.

- Please update the source data files according to the rearranged figures. Upload source data for EV figures as one zipped folder.

8) Synopsis: Every published paper now includes a 'Synopsis' to further enhance discoverability. Synopses are displayed on the journal webpage and are freely accessible to all readers. They include separate synopsis image and synopsis text.

- Synopsis image: Please provide a visual abstract as a high-resolution jpeg file 550 px-wide x (300-600)-px high to illustrate your article.

- Synopsis text: Please provide a short standfirst (maximum of 300 characters, including space) as well as 2-5 one sentence bullet points that summarise the paper as a .doc file. Please write the bullet points to summarise the key NEW findings. They should be designed to be complementary to the abstract - i.e. not repeat the same text. We encourage inclusion of key acronyms and quantitative information (maximum of 30 words / bullet point). Please use the passive voice.

9) As part of the EMBO Publications transparent editorial process (see our Editorial at

<http://embomolmed.embopress.org/content/2/9/329>), EMBO Molecular Medicine will publish online a Review Process File (RPF) to accompany accepted manuscripts. This file will be published in conjunction with your paper and will include the anonymous referee reports, your point-by-point response and all pertinent correspondence relating to the manuscript. Let us know if you want to remove or not any figures from it prior to publication. Please note that the Authors checklist will be published at the end of the RPF.

10) Please provide a point-by-point letter INCLUDING my comments as well as the reviewer's reports and your detailed responses (as Word file).

I look forward to reading a new revised version of your manuscript as soon as possible.

Yours sincerely,

Zeljko Durdevic

Zeljko Durdevic
Senior Editor
EMBO Molecular Medicine

*** Instructions to submit your revised manuscript ***

When preparing your revised manuscript, please refer to our guidelines: <https://link.springer.com/journal/44321/submission-guidelines#cms-Revised-submissions>. We perform an initial quality control of all revised manuscripts before re-review; failure to include requested items will delay the evaluation of your revision.

We require:

- 1) A .docx formatted version of the manuscript text (including legends for main figures, EV figures and tables). Please make sure that the changes are highlighted to be clearly visible.
- 2) Individual production quality figure files as .eps, .tif, .jpg (one file per figure). For guidance, download the 'Figure Guide PDF': <https://media.springernature.com/original/springer-cms/rest/v1/content/27825798/data/v1>.
- 3) A .docx formatted letter INCLUDING the reviewers' reports and your detailed point-by-point responses to their comments. As part of the EMBO Press transparent editorial process, the point-by-point response is part of the Review Process File (RPF), which will be published alongside your paper.
- 4) A complete author checklist, which you can download from our author guidelines. Please insert information in the checklist that is also reflected in the manuscript. The completed author checklist will also be part of the RPF.
- 5) Please note that all corresponding authors are required to supply an ORCID ID for their name upon submission of a revised manuscript.
- 6) It is mandatory to include a 'Data Availability' section after the Materials and Methods. Before submitting your revision, primary datasets produced in this study need to be deposited in an appropriate public database, and the accession numbers and database listed under 'Data Availability'. Please remember to provide a reviewer password if the datasets are not yet public.

7) For data quantification: please specify the name of the statistical test used to generate error bars and P values, the number (n) of independent experiments (specify technical or biological replicates) underlying each data point and the test used to calculate p-values in each figure legend. The figure legends should contain a basic description of n, P and the test applied. Graphs must include a description of the bars and the error bars (s.d., s.e.m.).

9) Our journal encourages inclusion of *data citations in the reference list* to directly cite datasets that were re-used and obtained from public databases. Data citations in the article text are distinct from normal bibliographical citations and should directly link to the database records from which the data can be accessed. In the main text, data citations are formatted as follows: "Data ref: Smith et al, 2001" or "Data ref: NCBI Sequence Read Archive PRJNA342805, 2017". In the Reference list, data citations must be labeled with "[DATASET]". A data reference must provide the database name, accession number/identifiers and a resolvable link to the landing page from which the data can be accessed at the end of the reference.

12) Author contributions: You will be asked to provide CRediT (Contributor Role Taxonomy) terms in the submission system. These replace a narrative author contribution section in the manuscript.

13) A Conflict of Interest statement should be provided in the main text.

14) Every published paper includes a 'Synopsis' to further enhance discoverability. Synopses are displayed on the journal webpage and are freely accessible to all readers. They include a short stand first (maximum of 300 characters, including space) as well as 2-5 one-sentences bullet points that summarizes the paper. Please write the bullet points to summarize the key NEW findings. They should be designed to be complementary to the abstract - i.e. not repeat the same text. We encourage inclusion of key acronyms and quantitative information (maximum of 30 words / bullet point). Please use the passive voice. Please attach these in a separate file or send them by email, we will incorporate them accordingly.

15) Include a Reagents and Tools Table as part of the Methods section, which can be downloaded from our author guidelines.

Graphs 800-1,200 DPI
Photos 400-800 DPI
Colour (only CMYK) 300-400 DPI"

*Additional important information regarding figures and illustrations can be found at
<https://media.springernature.com/original/springer-cms/rest/v1/content/27825798/data/v1>

***** Reviewer's comments *****

Referee #1 (Comments on Novelty/Model System for Author):

The authors have carefully considered the previous reviews and have made substantial improvements to the manuscript, including new neurochemical data (RT-qPCR for DAT/TH/VMAT2, striatal TH quantification) and an expanded analysis of both αMPT treatment in DAT-KO mice and Pifithrin-u treatment of A313V mice. These additions considerably strengthen the mechanistic and translational aspects of the study.

Referee #1 (Remarks for Author):

The authors have carefully considered the previous reviews and have made substantial improvements to the manuscript, including new neurochemical data (RT-qPCR for DAT/TH/VMAT2, striatal TH quantification) and an expanded analysis of both αMPT treatment in DAT-KO mice and Pifithrin-u treatment of A313V mice. These additions considerably strengthen the mechanistic and translational aspects of the study.

I only have one comment regarding the inclusion of data assessing dystonic features in A313V and DAT-KO mice. In the rebuttal letter, the authors provide interesting data from the pole reversal test, the foot fault test, and the tail suspension test to evaluate clasping. They also mention that they have "included these new results" with reference to the clasping data. The revised method description in the manuscript file also includes the methodological description of all these tests. Yet, I do not see the corresponding data in the main text or supplementary? Later in the rebuttal, the authors discuss that these experiments did not replicate previously reported dystonic phenotypes by Ng et al. and that, because the findings are negative and may depend on housing/experimental conditions, they prefer not to include them in the paper. The authors' caution in presenting "negative" or conflicting data on dystonia-related readouts in A313V and DAT-KO mice is understandable. Still, I would personally encourage that these results (or some of them) be included in the manuscript (maybe as supplementary data). In my view, these experiments are well chosen and informative. Their inclusion would promote scientific transparency, provide the community with concrete data relevant to replication efforts by others and to the broader challenge of modelling parkinsonism/dystonia in mouse models, and exemplify well-executed null results that are scientifically relevant both in and of themselves and in the context of reproducibility efforts.

Minor comment.

I would suggest that the authors also mention that they "do not observe severe akinesia, rigidity, or tremor after αMPT treatment" in A313V mice, as this is not implicit from the locomotor data alone

Referee #3 (Comments on Novelty/Model System for Author):

The technical quality of the study is high, supported by a broad and well-integrated set of biochemical, behavioral and pharmacokinetic analyses. The DAT-A313V mutation provides a patient-relevant, genetically precise mammalian model of DTDS that has not previously been established, medical impact is substantial.

Referee #3 (Remarks for Author):

Overall, the authors have been highly responsive to reviewer input and have substantially strengthened the manuscript with new data and mechanistic clarification. Suitable for publication.

1) Please implement the referee #1 suggestions.

We have implemented referee #1's suggestions and included the dystonia behaviors in the results section, Fig. EV1 d-g, and discuss these null findings in the results section with references to Ng et al., 2021, and Illiano et al., 2017, whose findings are in contrast to ours.

2) Please consider changing the current title to "Alpha-methyl-para-tyrosine and amphetamine ameliorate hyperactivity in a novel mouse model of Dopamine Transporter Deficiency Syndrome"

Thank you for this suggestion; we have changed the title to "Alpha-methyl-para-tyrosine and amphetamine ameliorate hyperactivity in a novel mouse model of Dopamine Transporter Deficiency Syndrome"

3) Figures: Please reduce number of figures by merging 2-3 current figures, e.g. Fig1 and 2, Fig3, 4 and 5, Fig6 and 7, Fig8 and 9, Fig10 and 11 and, Fig12 and 13. Please note that all figures should fit one page. Please check "Author Guidelines" for more information: <https://www.embopress.org/page/journal/17574684/authorguide#figureformat>

We have merged figures 3,4,and 5, and figures 8 and 9. We have also ensured that all figures adhere to the 7.08 width requirement and the maximum 9.08 length.

4) Author checklist: Please submit a complete checklist. <https://www.embopress.org/pb-assets/embo-site/EMBO%20Press%20Author%20Checklist-1642513524327.xlsx>

We have completed the author checklist and submitted it with the resubmission.

5)DONE In the main manuscript file, please do the following:

- DONE Please correct the citations of the EV figures to "Figure EV1" etc. and add callouts for Fig 2A, Fig 6F. Figures should be called out in a sequential order. DONE Currently Fig EV5B is called out before Fig EV4B, please correct. DONE Also, there are callouts for Fig. EV8d and Fig EV6 which do not exist, please correct.

We have corrected the citations of the EV figures and added callouts for Fig. 2a and Fig. 6f. We have ensured figures are called out in sequential order and removed callouts for non-existing figures.

- Remove data not shown on page 26 and in figure legends Fig 2,5,6, 7,8,9,11,13.

We have removed "data not shown" from relevant figures.

- Please include structured Methods section that includes a Reagents and Tools Table (should be uploaded as a separate file) followed by a Methods and Protocols section. More information on how to adhere to this format as well as downloadable templates (.docx) for the Reagents and Tools Table can be found in our author guidelines: <https://www.embopress.org/page/journal/17574684/authorguide#structuredmethods>

An example of a paper with Structured Methods can be found here:

<https://www.embopress.org/doi/full/10.1038/s44320-024-00037-6#sec-4>

We have included a Tools and Reagents checklist and a structured methods section.

- Indicate in legends exact n and exact p values, not a range, along with the statistical test used. To keep the figures "clear" some authors found providing an Appendix table Sx with all exact p-values preferable. You are welcome to do this if you want to.

We have ensured legends report exact n and p values and not a range along with the statistical tests used.

Rename "Conflict of interest" to "Disclosure and competing interests statement". We updated our journal's competing interests policy in January 2022 and request authors to consider both actual and perceived competing interests. Please review the policy <https://www.embopress.org/competing-interests> and update your competing interests if necessary.

We have renamed the section to "Disclosure and competing interests statement"

In data availability statement replace current text with "This study includes no data deposited in external repositories."

We have replaced the current text with ""This study includes no data deposited in external repositories."

Merge Funding with Acknowledgement.

We have merged funding with acknowledgements.

Please correct the reference citation in the reference list. Citations should be listed in alphabetical order. Currently Mir, A., et al is after Aguilar, J. I., et al. Where there are more than 10 authors on a paper, 10 will be listed, followed by "et al.". Also, please remove DOIs. DOIs should only be used for preprints and datasets that have not been published. Please check "Author Guidelines" for more information.

<https://www.embopress.org/page/journal/17574684/authorguide#referencesformat>

We have corrected the Mir citation and when there are 10 or more authors, we have added et al, and removed DOIs.

6)Appendix: Please move Appendix Methods to main Methods section. Add page numbers to table of contents and correct to "Appendix Figure S1" etc. and "Appendix Table S1".

We have moved the appendix methods to the methods section, added page numbers to the appendix, and corrected table and figure titles.

7) Source data:

- uploaded PDF versions of the cropping for westerns in figure 1 and 2Please upload high-resolution source data images for blots in Fig1 and 2. Currently these are pixelated.
- Correct panel labeling for Fig12 source data to b, c, d.
- Please update the source data files according to the rearranged figures. Upload source data for EV figures as one zipped folder.

We have uploaded PDF versions of the cropping for western blots in figures 1 and 2, corrected the panel labeling for Fig 10 source data, and updated the source data files according to the rearranged figures. We have also uploaded source data in one zipped folder.

8) Synopsis: Every published paper now includes a 'Synopsis' to further enhance discoverability. Synopses are displayed on the journal webpage and are freely accessible to all readers. They include separate synopsis image and synopsis text.

- Synopsis image: Please provide a visual abstract as a high-resolution jpeg file 550 px-wide x (300-600)-px high to illustrate your article.
- Synopsis text: Please provide a short standfirst (maximum of 300 characters, including space) as well as 2-5 one sentence bullet points that summarise the paper as a .doc file. Please write the bullet points to summarise the key NEW findings. They should be designed to be complementary to the abstract - i.e. not repeat the same text. We

encourage inclusion of key acronyms and quantitative information (maximum of 30 words / bullet point). Please use the passive voice.

We have generated a synopsis image and a synopsis .dox file upon resubmission.

9) As part of the EMBO Publications transparent editorial process (see our Editorial at <http://embomolmed.embopress.org/content/2/9/329>), EMBO Molecular Medicine will publish online a Review Process File (RPF) to accompany accepted manuscripts. This file will be published in conjunction with your paper and will include the anonymous referee reports, your point-by-point response and all pertinent correspondence relating to the manuscript. Let us know if you want to remove or not any figures from it prior to publication. Please note that the Authors checklist will be published at the end of the RPF.

There are no figures we wish to have removed prior to the publication of the RPF.

10) Please provide a point-by-point letter INCLUDING my comments as well as the reviewer's reports and your detailed responses (as Word file).

16th Jan 2026

Dear Dr. Salahpour,

We are pleased to inform you that your manuscript is accepted for publication and is now being sent to our publisher to be included in the next available issue of EMBO Molecular Medicine.

You may qualify for financial assistance for your publication charges - either via a Springer Nature fully open access agreement or an EMBO initiative. Check your eligibility: <https://link.springer.com/journal/44321/how-to-publish-with-us>

Zeljko Durdevic
Senior Editor
EMBO Molecular Medicine

>>> Please note that it is EMBO Molecular Medicine policy for the transcript of the editorial process (containing referee reports and your response letter) to be published as an online supplement to each paper. If you do NOT want this, you will need to inform the Editorial Office via email immediately. More information is available here: <https://link.springer.com/partners/embo-press/editorial-policies#Peer%20review>